

# On the development of a new prototype PTR-ToF-MS instrument and its application to the detection of atmospheric amines

Alexander Håland[1,2], Tomáš Mikoviny[1], Elisabeth Emilie Syse[1], and Armin Wisthaler[1,2]

[1]Department of Chemistry, University of Oslo, Oslo, Norway
[2]Centre for Biogeochemistry in the Anthropocene (CBA), University of Oslo, Oslo, Norway

*Correspondence to*: Armin Wisthaler (armin.wisthaler@kjemi.uio.no)

**Abstract**. We herein report on the development of a new prototype PTR-ToF-MS instrument that combines a hollow cathode glow discharge (HCGD) ion source with a focusing ion-molecule reactor (FIMR), which consists of a resistive glass drift tube surrounded by quadrupole rods. The new instrument configuration hybridizes the two main current commercial PTR-ToF-MS instrument designs. We provide a detailed technical description of the new analyzer and its optimized operational settings for detecting volatile amines via proton transfer reactions from hydronium ($H_3O^+$) or ammonium ($NH_4^+$) ions. We show that the

new prototype PTR-ToF-MS instrument is capable of monitoring rapid changes of sticky amines on the timescale of a few seconds and detects atmospheric variations of amines down to single digit pptV levels. Application examples given include the real-time monitoring of i) methylamine emitted from a *Chenopodium vulvaria* L. plant, ii) small alkylamines in ambient air on site of an agricultural research center (Senter for husdyrforsøk, Ås, Norway), and iii) an industrial amine (2-amino-2-methylpropan-1-ol, AMP) on site and downwind of a carbon dioxide ($CO_2$) capture test center (Technology Centre Mongstad

– TCM, Mongstad, Norway).



## 1 Introduction

Proton-Transfer-Reaction Mass Spectrometry (PTR-MS) is widely used in atmospheric sciences for the detection of non-methane organic gases (NMOGs) (de Gouw and Warneke, 2007; Hansel et al., 1995; Yuan et al., 2017). PTR-MS is a direct injection chemical ionization mass spectrometry (CIMS) technique. Direct injection means that the sample is directly

introduced into the analyzer without any precollection or pretreatment. The measurements are thus conducted in real time, with a frequency of up to 10 Hz. In its original and still most widely used configuration, PTR-MS uses gas-phase hydronium ions ($H_3O^+$) as chemical ionization (CI) reagent ions. $H_3O^+$ ions react with most NMOGs in non-dissociative proton transfer reactions (Hunt and Ryan, 1972; Lindinger et al., 1998). PTR-MS distinguishes itself from other CIMS techniques in the way that the ionization is effected in a small flow tube reactor, at reduced pressure (1-4 mbar) and under the action of an electrostatic

field (40-70 V cm$^{-1}$). The collisional drift of the ions at high velocities prevents them from getting hydrated and limits their residence time in the flow tube. Because of presence of the electrostatic field, the flow tube reactor is commonly referred to as the drift tube (DT). While the original PTR-MS instrument was equipped with a quadrupole mass spectrometer (QMS), state-of-the-art analyzers include a time-of-flight mass spectrometer (ToF-MS).

Currently, atmospheric composition researchers mainly use two types of PTR-ToF-MS instruments, which are manufactured

by two different producers. The instrument produced by Ionicon Analytik (Innsbruck, Austria) uses a hollow cathode glow discharge (HCGD) for producing $H_3O^+$ ions (Hansel et al., 1995; Müller et al., 2020). More details about this ion source are provided in the Methods section. In the conventional and widely used Ionicon instruments, the DT is composed of a stack of stainless steel (SS) rings, which are separated by insulating Teflon spacers. In the instrument produced by Tofwerk (Thun, Switzerland), also known as the VOCUS PTR-TOF or simply as "the VOCUS", $H_3O^+$ ions are formed in a water plasma

burning between two conical electrodes (Krechmer et al., 2018). In the VOCUS analyzer, the DT consists of a resistive glass tube surrounded by cylindrical quadrupole rods. The latter generate a focusing quadrupole field within the DT, which is thus commonly referred to as focusing ion-molecule reactor (FIMR).

Herein, we report on the development and application of a new prototype PTR-ToF-MS instrument that hybridizes the two commercial PTR-ToF-MS designs, combining a FIMR with an HCGD ion source. In a conventional stacked ring DT, the ion

swarm broadens as it drifts through the reactor and only a fraction of the ions is sampled through the exit aperture towards the mass analyzer. This loss in sensitivity can partly be compensated by placing an ion funnel at the end of the DT (Barber et al., 2012). Such an ion funnel is nowadays routinely installed in most Ionicon analyzers (Pugliese et al., 2020). In the FIMR, a focusing quadrupole field acts over the entire drift region, which minimizes ion losses and boosts the instrument sensitivity ($10^3$-$10^4$ cps ppbv$^{-1}$; Krechmer et al., 2018). We thus decided to build an instrument based on the FIMR design proposed by

Krechmer et al. (2018). Since the VOCUS ion source was not yet commercially available when we conceived the new instrument, we opted to couple the FIMR with the existing HCGD ion source. When the VOCUS ion source became available, we decided not to retrofit it onto our analyzer. This decision was mostly driven by chemical-analytical concerns that were based on our previous work on water plasma ion sources. The VOCUS ion source has the peculiarity that the entire water plasma effluent is introduced into the FIMR. Electron ionization processes in a water plasma form $O^+$, $H^+$, $H_2^+$, $OH^+$ and $H_2O^+$

cations, and $H_3O^+$ ions are only generated in subsequent reactions between these cations and $H_2O$ molecules. Our first concern was that this conversion to $H_3O^+$ ions might not be complete when ions enter the FIMR from the VOCUS ion source. NMOGs would thus also be partly ionized via electron transfer reactions, with the presence of non-protonated species complicating the interpretation of mass spectra. Our second and more serious concern was related to the fact that a water plasma effluent typically contains high densities of oxidants such as hydroxyl ($OH^\bullet$) radicals. Highly reactive NMOGs would undergo partial

oxidation within the FIMR, if exposed to $OH^\bullet$ radicals being introduced from the VOCUS ion source. This would result in important positive measurement artifacts for oxidized and highly oxidized compounds, even if only a very small and unnoticeable fraction ($\leq 0.1\%$) of highly reactive NMOGs was oxidized inside the FIMR. It is important to note that we have no experimental evidence that such processes do actually occur in the VOCUS analyzer. On the other hand, there was (and



still is) no detailed description or characterization of the VOCUS ion source in the scientific literature, which is why we

ultimately decided to install the well-characterized HCGD ion source in our PTR-ToF-MS instrument.

We will describe the new prototype PTR-ToF-MS instrument in the Methods section of this paper. In the Results section, we will describe its performance for the detection of volatile amines and show exemplary laboratory and field data. Amines were chosen as target analytes because of the increasing interest of the atmospheric chemistry community in this class of compounds (Ge et al., 2011). This interest is mainly driven by the important role that amines play in new particle formation (*e.g.*, Almeida

et al., 2013). Amines are also being emitted to the atmosphere from carbon dioxide ($CO_2$) capture plants. Since the slip of amines from $CO_2$ capture plants may result in the atmospheric formation of harmful substances (*e.g.*, nitrosamines and nitramines) (Nielsen et al., 2012), amines are receiving increased interest from the air quality and health community. Atmospheric amines are, however, particularly challenging to measure (Lee, 2022). This is on one hand caused by the fact that atmospheric levels are usually extremely low, requiring analyzers to be able to detect single-digit pptV levels. On the other

hand, amines are also known to be "sticky", meaning that they easily adsorb onto surfaces in the analyzer and in the inlet lines. We will show that our new PTR-ToF-MS instrument is capable of detecting single-digit pptV levels of amines in the atmosphere. We will also demonstrate that our new instrument has only minor memory effects for amines and is thus capable of monitoring rapid changes of amine levels in the air on the timescale of a few seconds.

## 2 Experimental

### 2.1 Instrument description

The new prototype PTR-ToF-MS analyzer that combines a conventional HCGD ion source with the more recently developed FIMR is shown in Figure 1.

The HCGD ion source and the processes occurring therein have been described in detail in the original PTR-MS publication (Hansel et al., 1995). Only minor modifications have been implemented since then and the current configuration has been

described by Mülller et al. (2020). Only the essentials are thus outlined here (Figure 1). A DC water plasma is generated inside a plasma source consisting of a hollow cylinder anode, a hollow cylinder cathode and an anode lens with an exit orifice. The entire effluent from this plasma source, which includes ions, electrons, radicals and mostly neutral $H_2O$ molecules, is introduced into a hollow cylinder electrode, which is commonly referred to as the "source drift region" or "source drift ring" (SDR). The SDR is a key component of the ion source set-up, because it is mostly here where $O^+$, $H^+$, $H_2^+$, $OH^+$ and $H_2O^+$

cations react with $H_2O$ molecules to form $H_3O^+$ ions. It is also here where free electrons recombine with cations and where $OH^\bullet$ and other radical species are destroyed on SS surfaces. Importantly, the water vapor and other neutral plasma effluents are pumped off from the SDR and thus not brought in contact with the sample gas that is introduced into the instrument further downstream. Only ions ($\geq$98% $H_3O^+$, with $O_2^+$ and $NO^+$ making up for the rest) are extracted via the SDR exit lens and injected into the drift region. The entire ion source set-up can be heated. Five parameters can be set in the HCGD ion source: i) the

source gas flow ($\Phi_{source}$), ii) the source temperature ($T_{source}$), iii) the plasma current ($I_{plasma}$), iv) the voltage between the anode lens and the SDR ($U_{SDR,in}$), v) the voltage between the SDR and the SDR exit lens ($U_{SDR,out}$). The specific settings used for the measurements reported herein will be reported below. The HCGD ion source has also been used for generating other types of CI reagent ions such as $O_2^+$ or $NO^+$ (Jordan et al., 2009) and $NH_4^+$ (Müller et al., 2020; Zhu et al., 2018). We operated the new instrument only in the $H_3O^+$ mode and in the $NH_4^+$ mode, since both of these CI reagent ions can be used for detecting amines

via non-dissociative proton transfer reactions (Zhu et al., 2018).

The FIMR has been described in detail by Krechmer et al. (2018). We also use a 10 cm long resistive glass tube, but with a slightly smaller diameter (8 mm OD, 6 mm ID). Four operational parameters can be set in the FIMR: the frequency ($f_{FIMR}$) and peak-to-peak amplitude voltage ($V_{pp, FIMR}$) of the RF applied on the quadrupole rods, the DC voltage applied across the ends of the glass tube ($U_{drift}$) and the pressure ($p_{drift}$) in the vacuum chamber in which the resistive glass tube is placed. The sample



105 gas is introduced into the FIMR via a custom-made passivated (Sulfinert®, Restek, Bellefonte, PA, U.S.A.) gas inlet lens (GIL). The GIL is placed between the SDR exit lens and the FIMR. Two additional voltages can be set relative to the GIL: i) the voltage between the SDR exit lens and the GIL ($U_{GIL,in}$) and ii) the voltage between the GIL and the upper end of the FIMR ($U_{GIL,out}$). The GIL is connected to a 15 cm long 1/16-in. OD Teflon PFA tube, which includes a T-piece to branch off part of the inlet flow into a digital pressure controller (Bronkhorst High-Tech B.V., Ruurlo, The Netherlands). This device keeps the

110 pressure in the Teflon inlet tube ($p_{inlet}$) (and thereby also $p_{drift}$) constant. The inlet line and gas inlet lens can be heated ($T_{inlet}$), which indirectly heats the FIMR. The GIL has a second port that is connected to a capillary leak through which a mass axis calibration gas (1,4-diiodobenzene; Sigma-Aldrich, Oslo, Norway) diffuses into the FIMR. The sample gas is pumped off through a 1 mm gap between the lower end of the FIMR and the skimmer lens at the entrance of the mass spectrometer. The skimmer is grounded and the voltage between the lower end of the FIMR and the skimmer ($U_{FIMR,out}$) can be set. All specific

115 settings used for the measurements reported herein will be reported below. One of our key strategies for reducing the instrumental time response was to maximize the flow through the low-pressure flow reactor. A dry multi-stage Roots pump (ACP40; Pfeiffer Vacuum, Asslar, Germany) pumps up to 0.80 slpm (standard liters per minute) through the FIMR. For the measurements presented herein, we set the flow rate to 0.55 slpm, which is a factor 5 higher than in commercial VOCUS instruments (Krechmer et al., 2018; Wang et al., 2020).

120 The mass analyzer used in our instrument is a quadrupole ion guide orthogonal acceleration reflectron ToF-MS (model HTOF; Tofwerk, Thun, Switzerland). The quadrupole ion guide (commonly referred to as the big segmented quadrupole, BSQ) was operated at an RF frequency of 4.4 MHz and an RF amplitude between 300 and 320 V. It is important to note that we operate the BSQ at relatively high RF frequencies to avoid strong mass discrimination effects in the $m/z$ 30 to 60 region, which is typically observed for commercial VOCUS analyzers (Krechmer et al., 2018; Wang et al., 2020).

125 **2.2 Laboratory test**

The first application test was carried out in the laboratory. We cultivated a Stinking Goosefoot (*Chenopodium vulvaria* L.) plant, which is a weed covered by minuscule bladder hairs. Upon mechanical stress, these bladder hairs release trimethylamine (TMA). The plant was placed in our laboratory next to the analyzer and mechanically stressed. The evolving TMA puff was monitored in real time by our new prototype PTR-ToF-MS instrument.

130 **2.2 Field tests**

A field test was carried out on campus of a research center for animal husbandry (Livestock Production Research Centre, Norwegian University of Life Sciences, Ås, Norway). The prototype PTR-ToF-MS analyzer was placed in a temperature-controlled enclosed car trailer. The trailer was placed close to large manure tanks where the animal excrements from the entire site are collected. The aim was to test the capability of monitoring atmospheric amines, with a focus on methylamine (MMA),

135 dimethylamine (DMA) and TMA. Air was sampled through a hole in the lateral wall of the trailer, using a 130 cm long, heated (100 °C), Sulfinert®-passivated, 1/4-in. OD SS tube pumped at 25 slpm. The instrument subsampled from this main inlet flow. A flow of 2.5 sccm (standard cubic centimeters per minute) of 1-3% ammonia ($NH_3$) in nitrogen (Nippon Gases Norge AS, Oslo, Norway) was continuously added at the tip of the main inlet for chemically passivating all wetted surfaces in the inlet and in the instrument itself (Roscioli et al., 2015; Zhu et al., 2018). Excess ammonia is known to displace amines adsorbed

140 onto surface sites (Ongwandee et al., 2005). The instrumental background was determined by overflowing the instrument subsampling flow with ambient air that had passed through a heated (350 °C) Pt/Pd catalyst. Animal excrements emit a plethora of NMOGs, which are known to produce very complex mass spectra in PTR-ToF-MS instruments (Kammer et al., 2020; Pedersen et al., 2021). We thus decided to operate our instrument using $NH_4^+$ as the CI reagent ion and to increase the collisional energy in the FIMR for suppressing $NH_4^+$ adduct formation and for dissociating $NH_4^+$ adducts. In such an operation

145 mode, only analytes with a proton affinity >204 kcal mol⁻¹ ( *i.e.*, the proton affinity of $NH_3$, Hunter and Lias, 1998) are



efficiently detected in their protonated form (Zhu et al., 2018). We thus refer to this operation mode as the $NH_4^+$-proton transfer ($NH_4^+$-PT) mode, which differs from the $NH_4^+$-adduct formation mode recently reported in the literature (Hansel et al., 2018; Zaytsev et al., 2019). In the $NH_4^+$-PT mode only a few classes of analytes such as amines and amides are ionized, making the PTR-ToF-MS mass spectra simpler to interpret. Another advantage of the $NH_4^+$-PT mode is that the instrumental response

does not exhibit any humidity dependence (<3% analyte ion signal change over the range of typical ambient humidities). Calibrations were performed in the field using a liquid calibration unit (LCU) (LCU-a, Ionimed Analytik GmbH, Innsbruck) (Fischer et al., 2013). An aqueous standard solution of MMA, DMA and TMA was prepared by diluting known volumes of commercial solutions of MMA (40±3% wt. in $H_2O$), DMA (40±2% wt. in $H_2O$), and TMA (45±3.5% wt. in $H_2O$) with Type 1 Milli-Q® water. In the LCU, the standard solution was nebulized/evaporated at a known liquid flow rate (2-10 µl min$^{-1}$) into

a known flow (typically 1 slpm) of NMOG-free air. The operational parameters used for the measurements at the agricultural research center were as follow: $\Phi_{source}$ = 6 sccm (1-3% $NH_3$ in $N_2$; Nippon Gases Norge AS, Oslo, Norway); $I_{source}$ = 4 mA; $T_{source}$ = 100 °C; $T_{inlet}$ = 100 °C; $U_{SDR,in}$ = 234 V; $U_{SDR,out}$ = 200 V; $U_{GIL,in}$ = 78 V; $U_{GIL,out}$ = 50 V; $U_{drift}$ = 450 V; $U_{FIMR,out}$ = 40 V; $p_{drift}$ = 1.95 mbar, $f_{FIMR}$ = 5.5 MHz, $V_{pp, FIMR}$ = 350 V. The FIMR was operated at relatively high RF frequencies to avoid strong mass discrimination effects in the $m/z$ 30 to 60 region.

Another field test was conducted at and near the Technology Centre Mongstad (TCM; Mongstad, Norway). TCM is one of the world's largest test centers for amine-based $CO_2$ capture (de Koeijer et al., 2011). The measurements were carried out during a test campaign with a solvent containing 2-amino-2-methylpropan-1-ol (AMP) (Languille et al., 2021). The instrument was calibrated with the LCU using an aqueous solution of AMP, which was prepared by diluting a weighted amount of the pure compound (≥99%; Sigma-Aldrich, Oslo, Norway) in Type 1 Milli-Q® water. The first aim was to demonstrate that the new

instrument is capable of detecting leaks of the solvent amine in real time. For this purpose, a gas mixture containing 25 ppmV of AMP was deliberately released into the air at approximately one meter distance from the instrument inlet using a HovaCAL® (model digital 211-MF; IAS GmbH, Oberursel, Germany) gas generator. During the test campaign at TCM, the treated flue gas emitted from the amine plant into the atmosphere contained ~1 ppmV of AMP. The second aim was to demonstrate that the new highly sensitive PTR-ToF-MS analyzer is capable of detecting such emissions in highly diluted form,

*i.e.* at pptV levels, downwind of the industrial site. For this purpose, the instrument trailer was placed ~4.4 km north of TCM during a period with predominantly southerly winds. For the measurements at and near the industrial site, the prototype PTR-ToF-MS analyzer was operated in the conventional $H_3O^+$ mode. Some operational parameters were thus modified as compared to the first field campaign: $\Phi_{source}$ = 6 sccm ($H_2O$); $I_{source}$ = 4.5 mA; $U_{SDR,in}$ = 377 V; $U_{SDR,out}$ = 202 V; $U_{GIL,in}$ = 89 V; $U_{drift}$ = 400 V.

**3 Results and Discussion**

**3.1 Laboratory testing: detection of biogenic amines**

Figure 2 shows the time evolution of TMA in laboratory air when a *Chenopodium vulvaria* L. plant was mechanically stressed by tipping a leaf with a pen. The stress resulted in burst of TMA and the new prototype PTR-ToF-MS instrument detected a short-term spike in TMA with a peak width (FWHM) of 16 seconds. This was our first experimental evidence that the new

prototype instrument is capable of monitoring rapid changes of amine levels in the air on the timescale of a few seconds. The TMA volume mixing ratio did not drop to pre-spike levels, because the amine was released into and sampled from laboratory air. We did not make any further attempts to quantify TMA emission rates.



### 3.2 Field testing: detection of agricultural amines

After the first successful laboratory tests, the instrument was taken to the field. The goal was to measure small alkylamines in

ambient air on campus of an agricultural research center. As mentioned in the Methods section, the prototype PTR-ToF-MS instrument was operated in the $NH_4^+$-PT mode.

Figure 3 shows the calibration curves for MMA, DMA and TMA, as obtained from a calibration carried out in the field. The calibration curves show the normalized ion signal response (in *normalized counts per second*, ncps) as a function of amine volume mixing ratios in the 0 to 10 ppbV range. The three amines were detected in their protonated form at *m/z* 32.049, 46.065

and 60.081, respectively. The signals were normalized to the average $NH_4^+$ (*m/z* 18.034) signal observed over the course of the experiment. The protonated amines were detected with a mass resolving power (m/Δm) of 3700, 4100, and 4250, respectively. The instrumental response factors (sensitivities) for MMA, DMA and TMA were 114, 156, and 126 ncps ppbV$^{-1}$, respectively. The small spread in these values confirms that no significant mass discrimination effects occur in the *m/z* 32 to 60 range. We note that we usually set the upper *m/z* limit in the TOF-MS to ~500. Reducing the upper *m/z* limit to 100 would

increase the sensitivity by a factor of 2.2 (higher duty cycle). We also note that the measurement set-up (LCU) suffered from a background from previous amine calibrations, which explains the relatively high offset in the linear regression curves.

The instrumental response factors of the prototype PTR-ToF-MS analyzer are 1-2 orders of magnitude lower than those reported for commercial VOCUS analyzers (Krechmer et al., 2018; Wang et al., 2020). This is a result of a combination of various factors. First, we generally observe lower instrumental sensitivities when operating PTR-MS instruments in the $NH_4^+$-

PT mode (Zhu et al., 2018). Second, our prototype instrument has a relatively large gap between the FIMR and the skimmer (to increase the flow through the reactor and thereby minimize memory effects), which results in increased ion losses at the end of the FIMR. Third, we apply a relatively low RF amplitude to the FIMR, because we noticed that at the higher $V_{pp}$ used in previous studies (Krechmer et al., 2018; Wang et al., 2020), $O_2^+$ ions efficiently form within the FIMR. A high $O_2^+$ signal on *m/z* 31.989 severely compromises the detection of MMA at *m/z* 32.049. Fourth, we also apply a relatively low RF amplitude

to the BSQ for avoiding strong mass discrimination effects in the *m/z* 32 to 60 region. High instrumental response factors exceeding 1000 cps ppbv$^{-1}$ are of course desirable, but our priority in this first deployment was to generate more easily analyzable and interpretable data.

Prior to starting the measurements, we carried out another field test. Indole was among the analytes of interest for this field study (even though it was later only sporadically detected at very low levels), and laboratory tests had revealed that this amine

is particularly sticky. The headspace from a vial containing indole was thus shortly blown towards the instrument inlet and the protonated indole signal at *m/z* 118.065 was monitored at 1-second time resolution. Figure 4 shows the time series of indole as observed during this experiment. The initial volume mixing ratio on the order of 100 ppbV decreased by two orders of magnitude within 4 seconds and background levels were reached within ~40 seconds. We are fully aware that this instrument test deviates from the formal experimental procedure for determining an instrumental response time. However, these data do

show the performance of the analyzer under field conditions and allow us to conclude that the new prototype instrument is indeed capable of monitoring rapid changes of sticky amines on the timescale of a few seconds.

Figure 5 shows the time series of small alkylamines as measured in ambient air on site at the agricultural research center over a period of three days. It is important to note that direct introduction CIMS methods do not resolve isomeric species. DMA has only one additional isomer (ethylamine, EA), while TMA has three isomers (ethylmethylamine, propylamine,

isopropylamine). Previous work (Schade and Crutzen, 1995 and references therein) found trace levels of both EA and DMA in husbandry emissions, which is why we assign the *m/z* 46.065 ion signal to both of these compounds. The same literature reports that TMA is the dominant C$_3$-isomer, which is why we assign the *m/z* 60.081 ion to TMA only. We observed a clear diurnal pattern in the time profile of all three species, with maxima occurring during nighttime when the planetary boundary



layer is low. It is worth highlighting the time profile of DMA/EA, as it demonstrates the capability of our new prototype PTR-ToF-MS analyzer to monitor atmospheric variations of amines in the single to double digit pptV level range.

### 3.3 Field testing: detection of an industrial amine

The field tests at and near the $CO_2$ capture plant were carried out with the instrument being operated in the conventional $H_3O^+$ mode. The calibration curve (Figure 6) shows the *m/z* 90.091 normalized ion signal response (in *ncps*, normalized to the average $H_3O^+ + H_3O^+(H_2O)$ signal observed over the course of the calibration experiment) as a function of the AMP volume mixing ratio. The mass resolving power $(m/\Delta m)$ for this ion signal was 3700.

The calibration data show that in the conventional $H_3O^+$ mode, our prototype PTR-ToF-MS instrument does generate an instrumental response that exceeds $10^3$ ncps ppbv$^{-1}$. This is still on the lower end of what has been reported for commercial VOCUS analyzers. As mentioned above, this is owed to our choice to suppress $O_2^+$ formation in the FIMR by applying a low $V_{pp,FIMR}$ and to increase the FIMR flow by having a relatively large gap between the FIMR and the skimmer.

The first test carried out at TCM was to demonstrate the new instrument is capable of detecting amine solvent leaks in real time. A gas mixture containing ~25 ppmV AMP was deliberately released into the air at approximately one meter distance from the instrument inlet. The test was carried out during strong and turbulent wind conditions, which are typical for the Norwegian west coast. As seen in Figure 7, single to double digit ppbV levels of AMP were observed in real time. Notably, the instrument response is immediate and the signal quickly returns to baseline levels once the artificial leak has been closed.

The second test was to demonstrate that the new highly sensitive PTR-ToF-MS analyzer is capable of detecting ultra-trace levels of industrial amines in the atmosphere. During a test campaign, a steady level of ~1 ppmV AMP was released to the atmosphere by the amine plant at TCM. During a period with prevailing southerly winds, the car trailer with the prototype PTR-ToF-MS was placed ~4.4 km north of TCM. The measurements were carried out continuously and remotely for a period of 7 days. Figure 8 shows a 5-hour time series during which the plume emanating from TCM was intercepted. We note that at 1-minute signal integration signal variations of less than 1 pptV can be observed.

### 4 Conclusion

The PTR-MS technique was developed in the mid-1990ies at the University of Innsbruck. Initial method improvements focused on using different mass analyzers (*e.g.*, Blake et al., 2004; Prazeller et al., 2003). The ion source and drift tube only underwent minor geometrical changes over a period of two decades. The past five years have, however, seen major developments in the drift tube and ion source design (*e.g.*, Breitenlechner et al., 2017; Krechmer et al., 2018). We have herein presented one additional instrument variation, in which the well-characterized and "clean" HCGD ion source originally described by Hansel et al. (1995) is combined with the recently developed and highly sensitive FIMR design by Krechmer et al. (2018). Exemplary data have been provided to demonstrate the capabilities of this new instrument in monitoring atmospheric amines at single digit pptV levels. We have also shown that the new instrument has only minor memory effects for sticky amines, making it capable of monitoring rapid changes of amine levels in the air on the timescale of a few seconds. Further work is, however, required for achieving a time response of 1 second or less. This could be achieved by further optimizing the gas inlet or by heating the FIMR to ≥100 °C. We have also made an argument that maximizing the instrument sensitivity should not be the only focus of PTR-MS instrument development and optimization. Minimizing potential neutral and ionic side reactions in the DT, reducing sample memory effects and eliminating mass discrimination effects over the *m/z*-range of interest are important for generating easily analyzable, interpretable and quantifiable mass spectra. This is particularly important for the non-expert user. The PTR-MS method is being rapidly developed in various directions, and different types of PTR-MS instruments and operation modes are nowadays being used for different types of applications. The PTR-ToF-MS presented herein has been optimized for the detection of atmospheric amines, which are currently in the research focus of our



group. In future work, we will show how we have used the new prototype PTR-ToF-MS instrument for quantifying amine

emission rates from animal husbandry and for investigating the atmospheric impact of amine emissions from $CO_2$ capture.

*Data availability.* The data used in this paper are available from the corresponding author upon request.

*Author contribution.* TM and AW conceived the instrument. TM and AH built and optimized the instrument. AH carried out

the lab and field measurements, with help from EES and logistical support from TM. AH analyzed all data. AH drafted the
manuscript. AW wrote the final manuscript. All authors commented and accepted the final version of the manuscript.

*Competing interests.* The authors declare that they have no conflict of interest.

*Acknowledgements.* We thank the Senter for husdyrforsøk (SHF) at the Norwegian University of Life Sciences (NMBU) for
hosting our new prototype instrument. Irma Caroline Oskam and Birgitte Mosveen from SHF are acknowledged for their
hospitality and logistical support. We also thank the Technology Centre Mongstad (TCM) and in particular Audun Drageset
and Øyvind Ullestad for logistical support. Baptiste Languille is acknowledged for logistical support of the field measurements
at TCM. Special thanks go to Marc Gonin from Tofwerk for sharing his early ideas on quadrupole-guided resistive glass tubes

and to Tofwerk for providing a prototype FIMR. We apologize that it took so long to make something out of it.

*Financial support.* AH received a doctoral fellowship from the Centre for Biogeochemistry in the Anthropocene (CBA) at the
University of Oslo. AW acknowledges financial support for the acquisition of the HTOF from the University of Oslo via his
start-up grant. TCM financially supported the measurements at and near its site through the ARM (Amine Research and

Monitoring) project.





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




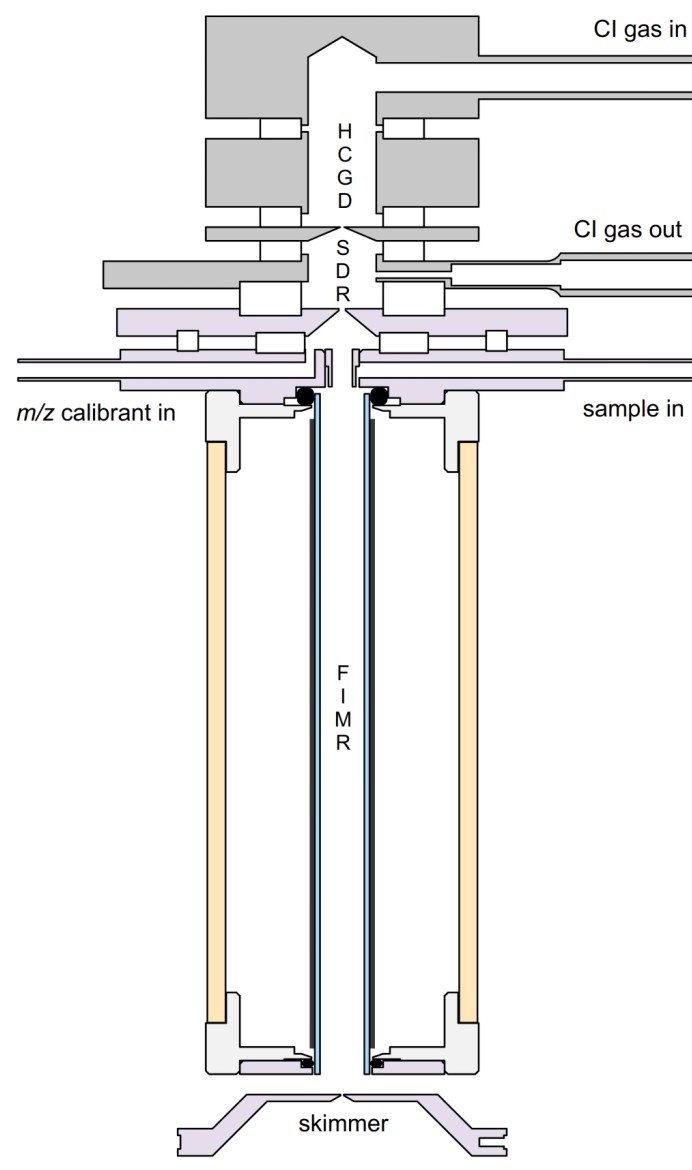

**Figure 1. Schematic drawing of the new prototype PTR-ToF-MS analyzer. The instrument consists of a hollow cathode glow discharge (HCGD) ion source, a source drift region (SDR) and a focusing ion-molecule reactor (FIMR), which is a resistive glass**
**drift tube surrounded by cylindrical quadrupole rods.**

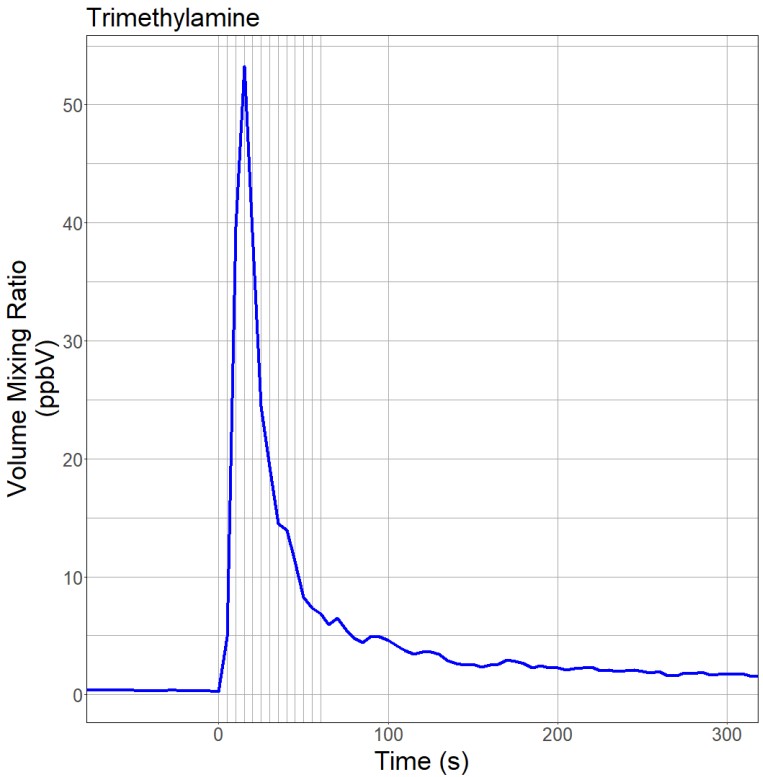

**Figure 2. Time series of trimethylamine as measured in laboratory air when a *Chenopodium vulvaria* L. plant was mechanically
stressed by tipping a leaf with a pen.**




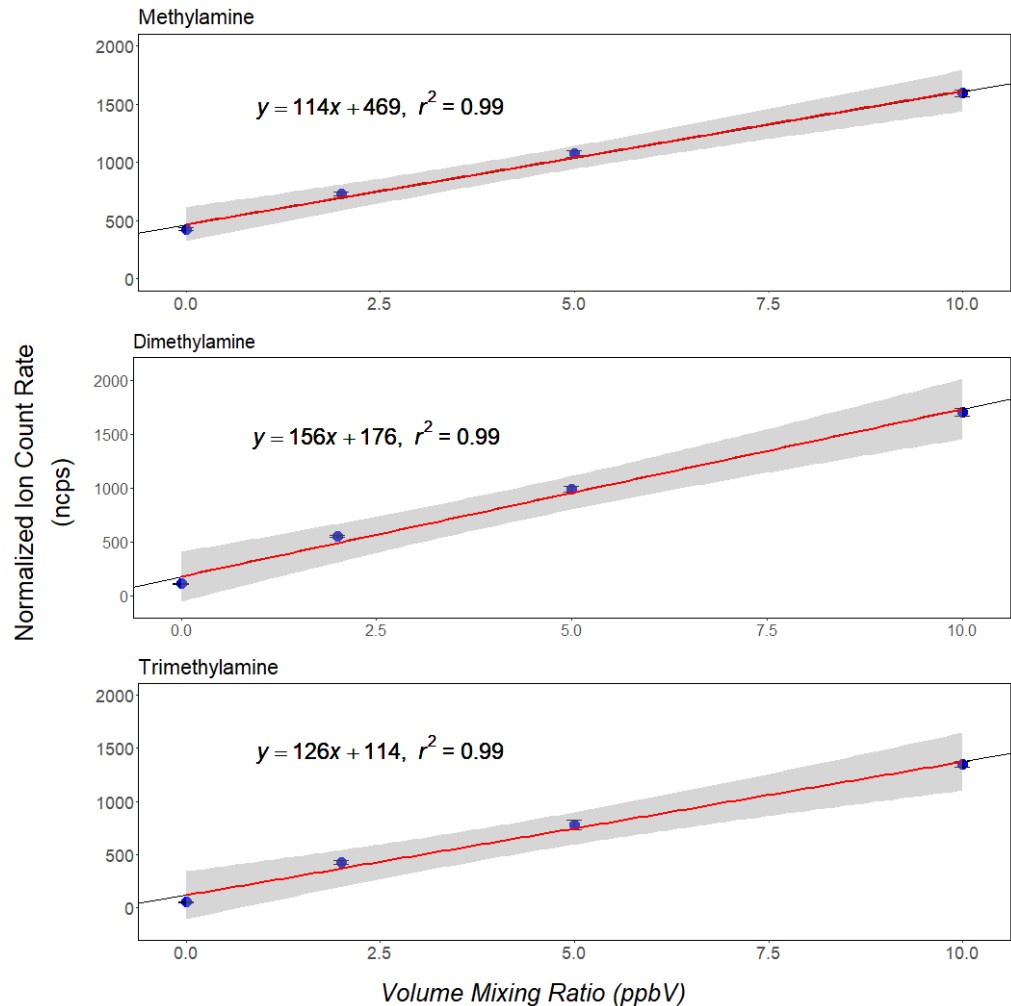

**Figure 3. Calibration curves for methylamine (upper panel), dimethylamine (middle panel), and trimethylamine (lower panel). The**
**PTR-ToF-MS instrument was operated in the NH$_4^+$-PT mode (see text).**





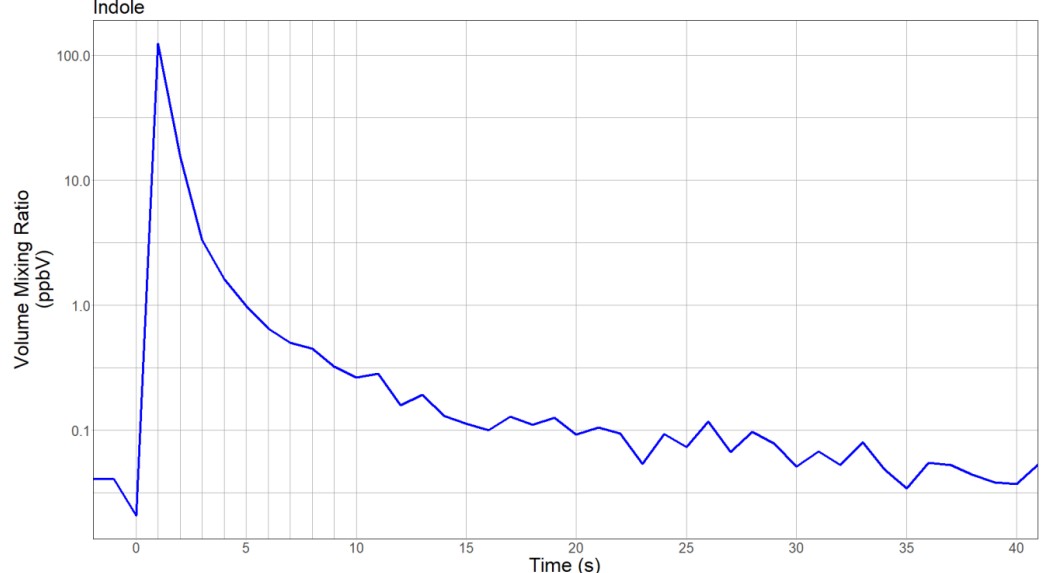

**Figure 4. Time evolution of the observed indole VMR when the headspace from a vial containing indole was shortly blown towards the PTR-ToF-MS instrument in its field configuration.**



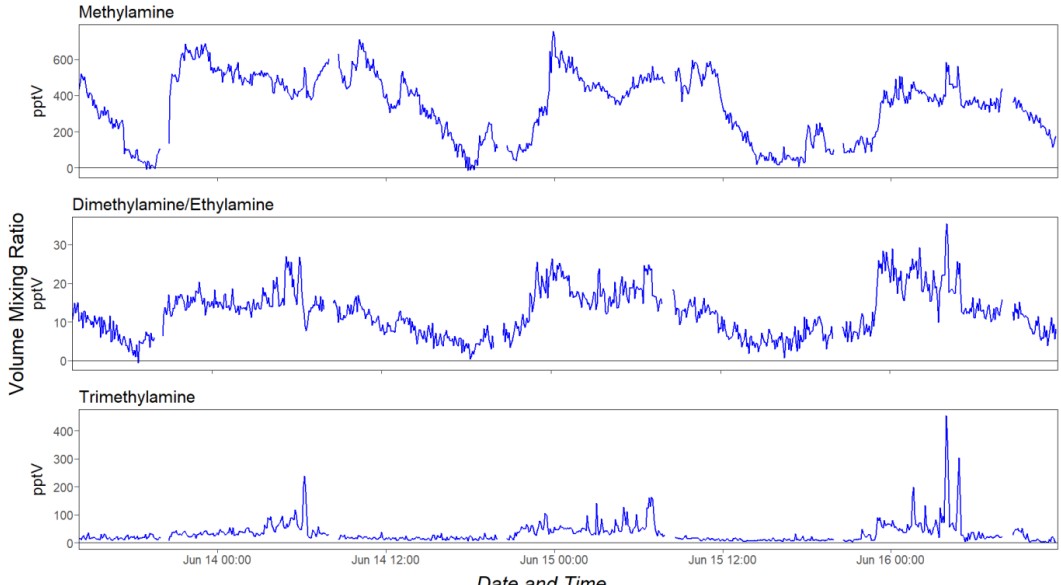

**Figure 5. Time series of methylamine (upper panel), dimethylamine/ethylamine (middle panel), and trimethylamine (lower panel) as observed in ambient air on site of an agricultural research center.**

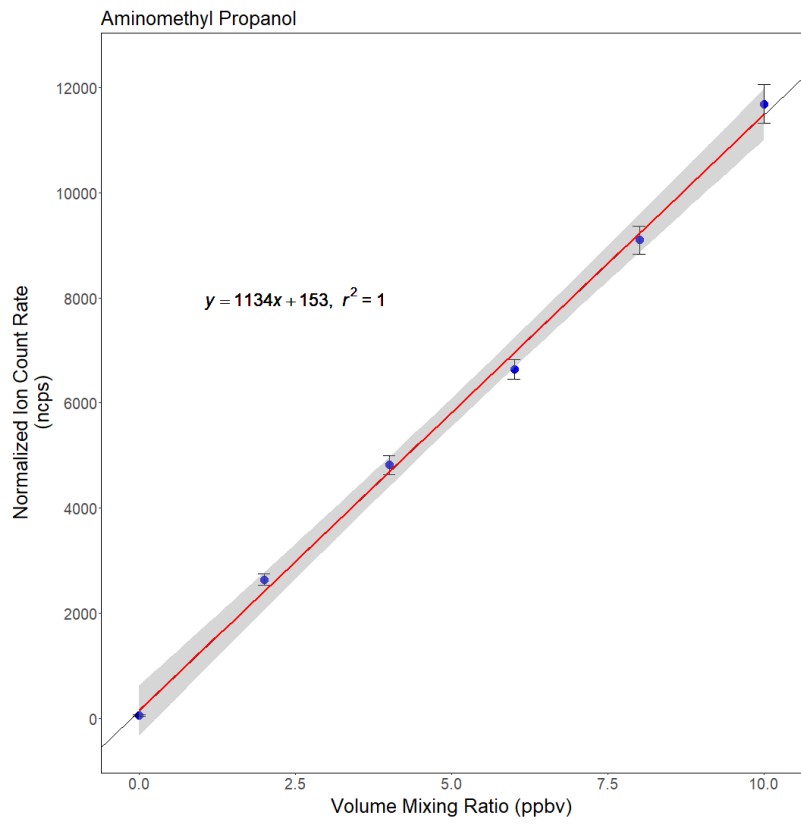


**Figure 6: Calibration curve for 2-amino-2-methylpropan-1-ol (AMP). The PTR-ToF-MS instrument was operated in the H₃O⁺ mode (see text).**



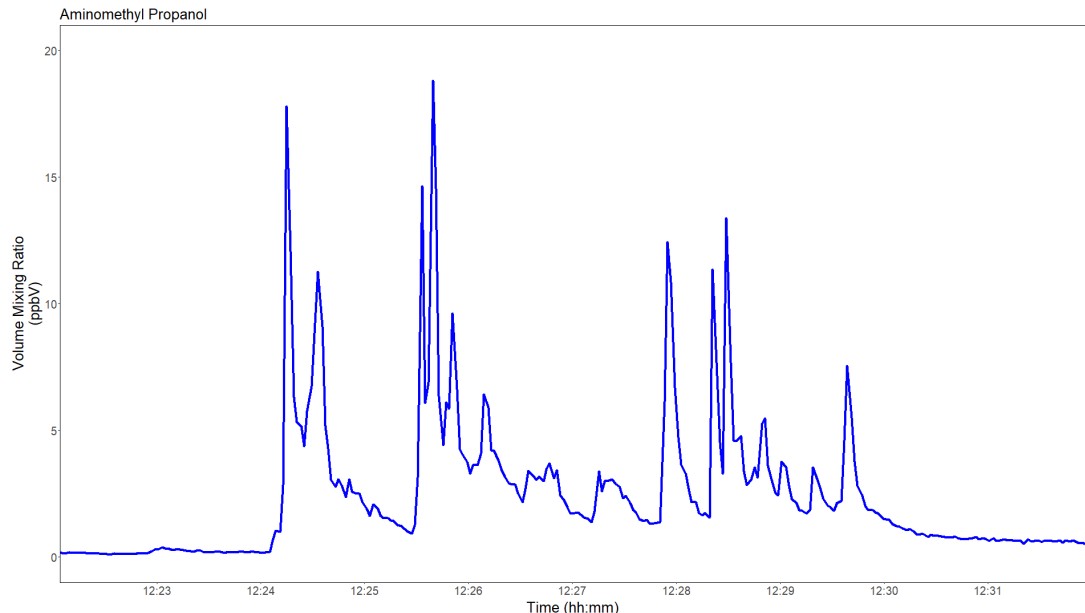


**Figure 7: Time series of 2-amino-2-methylpropan-1-ol (AMP) as measured when ~25 ppmV of this industrial amine were deliberately released into the air at approximately one meter distance from the instrument inlet.**




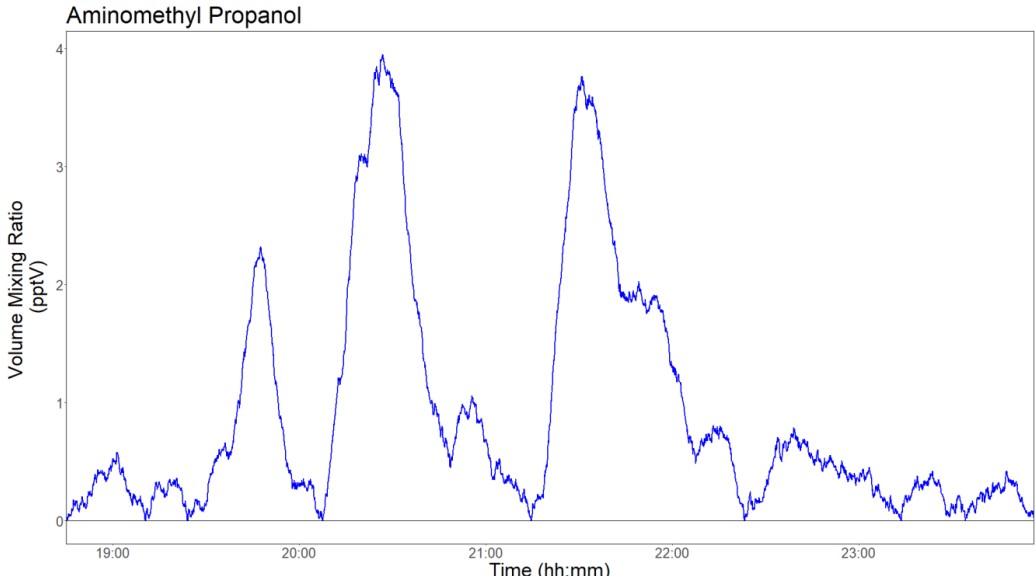

**Figure 8: Time series of 2-amino-2-methylpropan-1-ol (AMP) as measured at Sande, 4.4 km downwind from the TCM $CO_2$ capture plant. A steady level of ~1 ppmV of AMP was released from the stack during the measurement period, which resulted in single-digit pptV enhancements of AMP at the downwind site.**
