# Peer review of "On the development of a new prototype PTR-ToF-MS instrument and its application to the detection of atmospheric amines"

_Atmospheric Measurement Techniques, 2022_

## Author Response (AR1)

We thank referee #1 for having carefully read our manuscript and for making very valuable comments and suggestions. Here is how these were addressed:

*For measuring sticky compounds present at low mixing ratios such as amines, instrument and inlet background/zero determination is critically important. In order to properly judge the prototype instrument, further information on both instrument and inlet background are required. For instance, how frequently and for what length of time were zeros performed for the ambient measurements? How reproducible/stable were the backgrounds? Did relative humidity changes influence the background (particularly the inlet background)? Seeing a time series showing both ambient and background data would also support the claims about the short response time and the reduction in memory effects. Although the paper focuses on the instrument and not the inlet per se, information on the inlet background and response is necessary for the reader to critically evaluate the ambient measurements.*

**1: We fully agree that background/zero determination is a critically important aspect. In the revised manuscript, we describe how frequently and for what time length zeros were performed (lines 145-146, 175-176). We also add two figures in the Supplement showing the low and stable instrumental background for DMA, TMA and AMP. When reanalyzing the data, we noticed that the variable MMA background (measured at $m/z$ 32.049) correlates with the $O_2^+$ signal (measured at $m/z$ 31.989). It turned out that also most of the $m/z$ 32.049 signal variations in ambient air were caused by variations in the $O_2^+$ signal. We thus have to withdraw the MMA data in the revised version of the manuscript.**

*Given that this is an instrument paper, detection limits, precision, and accuracy should be reported.*

**2: Yes, this was also a major omittance. In the revised manuscript, we are reporting all these figures of merit (lines 160-161, 166-167, 202, 228, 230, 245, 259).**

*The time resolution of the measurements should be more clearly presented particularly for the data presented in figures 5, 7, and 8. Without this information, it is challenging for the reader to adequately judge instrument performance.*

**3: The time resolution is now reported in the respective figure captions and in the main text.**

*Several steps were taken to improve response time (increased flow through the flow reactor heated lines, NH3 addition). If available, I think it would be beneficial to include information on the relative impacts of these different steps. Steps such as heating lines to 100 degrees Celsius can be challenging in certain deployments and NH3 adds additional complexity (and corrosion concerns). It would be beneficial to the community to understand which practices are the most critical for response time.*

**4: Increasing the flow through the drift tube was the most critical step. This is now further emphasized in the manuscript: "Our key strategy for reducing the instrumental time response was to maximize the flow through the low-pressure flow reactor. A dry multi-stage Roots pump (ACP40; Pfeiffer Vacuum, Asslar, Germany) pumps up to 0.80 slpm (standard liters per minute) through the FIMR. For the measurements presented herein, we set the flow rate to 0.55 slpm, which is a factor 5 higher than in commercial VOCUS instruments (Krechmer et al., 2018; Wang et al., 2020). We observed that increasing the reactor flow from 0.05 to 0.50 slpm reduced the $1/e^2$ decay time of a 20 pptv trimethylamine (TMA) signal from ~60 to ~5 seconds." The other two measures (heating the lines, addition of $NH_3$) were unfortunately not investigated in a systematic way and were implemented based on our previous experience with measuring amines in atmosphere simulation chambers.**

*I ask the authors to consider adding examples of ambient mass spectra, particularly around the ions of interest, for the NH4+ and H3O+ modes. This would provide justification for the claims about simplifying interpretation.*

**6: We are withdrawing the claim about simplifying interpretation, at least when small alkylamines are the target compounds. While the overall mass spectrum becomes indeed simpler in the NH4+ mode, protonated DMA and TMA are easy to detect in a medium resolution mass spectrum both in the H3O+ and in the NH4+ mode. Our original assumption was that a lower O2+ background in the NH4+ mode would improve the detection of MMA, but a more careful reanalysis of the data revealed that this is not the case. Please also see comments #1 and #10 for more details.**

***Given that the applications described are amine measurements, the introduction should include a brief summary of the various techniques that have been used for amines rather than just focusing on comparisons to the VOCUS. Specific advantages relative to those measurements should also be detailed.***

**7: The paper is conceived as a description of new PTR-MS prototype instrument, and the introduction puts it into that context. We feel that an additional paragraph on different amine detection technologies. would disrupt the reading flow and not really fit into the introduction. We do however agree that additional direct injection CIMS methods should be mentioned and have thus included one extra sentence and several references: "Direct injection CIMS is thus the method of choice for measuring pptv levels amines in the atmosphere in real time, and a series of instruments have recently been developed for measuring amines (Sellegri et al., 2005; Hanson et al., 2011; Yu et al., 2012; You et al., 2014; Zheng et al., 2015; Yao et al., 2016; Wang et al., 2020; Pfeifer et al., 2020; Lee 2022)."**

***Please fix the section numbering (there are two section 2.2)***

**8: Done. Thanks for spotting this.**

We also thank referee #2 for having carefully read our manuscript and for her/his valuable comments and suggestions. Here is how we addressed these:

*In the introduction from around line 51-65 there is a verbal comparison of the HCGD ion source to the VOCUS source. There are several lines here that suggest potential issues with the VOCUS source, that are not supported by data, meant to indicate that these are issues with the VOCUS ion source. It seems inappropriate to criticize the VOCUS for theoretical issues which are not investigated within this work, nor are necessary to understand this manuscript. This seems inappropriate and I believe this section should be removed.*

**9: In our opinion, it is appropriate to mention potential problems of an existing instrument in the introduction of a chemical-analytical paper. We provide solid scientific arguments, explicitly state that these are only potential problems and do not use discrediting words. In our opinion, it should be allowed to raise concerns over an existing ion source design and make the scientific community aware of these. However, we do not insist on including this paragraph. If the Editor also thinks this is inappropriate, we simply remove these statements.**

*In general, this paper feels like it is lacking in the detail necessary to make it a strong addition to instrumental papers on the NH4+ techniques that are being used. Perhaps I missed it on my read throughs, but it doesn't seem like the authors even give details on what ion source fuel was used to generate NH4+, was it a water solution containing NH3? In terms of the ion generation more details on how clean the spectra are with respect to potential primary ions is needed, how effectively do the conditions identified/used reduce the NH3-NH4+ of NH3-H2O+ ions which may lead to alternative products that would complicate the spectra. Showing a spectrum and the relative abundance of these ions would be beneficial. Additionally, for any calibrants that were measured perhaps the authors could give a comprehensive list of where else in the spectra ions were observed. Is there any fragmentation occurring, or unintended products observed. Do you see both adduct and proton transfer products.*

**10: This paper is actually not intended to be "a strong addition to instrumental papers on the NH4+ techniques". The focus is the new instrumental design, which combines an HCGD ion source with an FIMR. We used this new instrument for measuring amines, which can by achieved by either operating in the H3O+ or NH4+ mode. We have removed the paragraphs/sentences that suggest the latter is more suitable for measuring small alkylamines.**
The source gas in the ammonia mode is described in lines 156 and 157 of the original manuscript: $\Phi_{source}$ = 6 sccm 1-3% $NH_3$ in $N_2$. In the revised manuscript, we provide a slightly modified and hopefully clearer description of the operational settings of the ion source (lines 148-151).
In the revised manuscript, we also list all additional ions with a relative abundance >1% of NH4+ (lines 150-151). We do not think it is necessary to add an extra figure for conveying this information.
We also explicitly state that [M-H]+ ions are the only additional product ions observed for the investigated amines (lines 196-197).

*Without an actual comparison to a VOCUS ion source it is difficult to understand if this HCGD source coupled to the FIMR TOF is indeed an advantage over the commercially available VOCUS package. Do the authors have any data that can serve as a point of comparison?*

**11: Unfortunately, we do not own a commercial VOCUS instrument and are thus not able to make such a comparison. We also want to emphasize that this paper should not be interpreted in a competitive context. We do not want to present a "product" that is better than existing commercial analyzers; we simply present a prototype instrument that is new and different.**

*It is clear that this instrument functions well, but the package is not commercially available and as far as I understand you cannot purchase a VOCUS with a HCGD, nor are the design plans for the HCGD publicly available to my knowledge, so I am not sure how useful this work*

*is if the instrument is a specialized one-of-a-kind instrument. Am I missing something, and this would in fact be a system that could be reproduced easily or purchased?*

**12: This is a comment we do not understand. The scientific literature (and in particular AMT) is full of descriptions of "specialized one-of-a-kind" instruments, i.e. instrument prototypes that are only used by one research group. Such papers mainly serve as a reference for future publications, in which the data collected with these new analyzers are presented. Another motivation is that other groups could build the same or a similar instrument. Again, there is no commercial motivation behind our work.**

*There are clearly data gaps in the time series shown in figure 5 which are possibly instrument zeros, it would be helpful to see the zero data in these figures, or at least an inset of data showing the zeroing. The same is desired for figures 7 and 8, where without the zeros it is impossible to determine if these features are driven by changes in instrument backgrounds or real ambient observations.*

**13: We now provide two additional figures in the Supplement that include the zero data.**

*In line 245 the authors point to the potential interpretation of 1 ppt signal variation on a 1-minute signal. Without knowledge of the instrumental detection limits, which are currently missing from this work it is unclear if you can even interpret that information. It is possible that the 1ppt variations are simply due to unstable instrument conditions. Showing background periods associated with that data and the experimentally determined detection limits during that measurement period is necessary to imply observations are interpretable on that level.*

**14: This is correct. We now include a statement on the detection limit and give an additional figure in the Supplement showing the variability of the instrumental background. "The instrumental background was low and stable (0.6 ± 0.2 pptV) throughout the measurement period (see Figure S2), which resulted in a detection limit ($3\sigma$, 5-min signal integration) of 0.5 pptV for AMP."**

*Specific comments:*

- *Line 31, rewrite to read "Because of the presence of an electrostatic field…"*
- *Line 45, ions is are sampled*
- *Line 67, the word exemplary is used here to describe laboratory and field data but it is nonquantitative and should be removed.*

**15: We have adopted two of the suggested changes. We think that "..fraction of the ions is sampled" is correct.**

---

## Author Response (AR2)

Dear Glenn,

Thanks for your feedback. We have made the following changes:

1) As suggested.
2) We included one mass spectrum in the Supplement (Figure S1) which should address the request made by reviewer #2 (reagent ions and "impurity ions"). Reviewer #1 requested to include an ambient mass spectrum to justify the claims about simplifying interpretation. We have however withdrawn this claim and we think it really would not add anything if we plotted the spectrum around $m/z$ 46 (DMA) and $m/z$ 60 (TMA). In addition, we actually did not measure in the conventional $H_3O^+$ mode during the first study, meaning that we cannot compare the mass spectra obtained in the two modes ($NH_4^+$ and $H_3O^+$).
3) Changed.

I hope this will work.

Armin